# The Detection System for a Danger State of a Collision between Construction Equipment and Workers Using Fixed CCTV on Construction Sites

**DOI:** 10.3390/s23208371

**Published:** 2023-10-10

**Authors:** Jaehwan Seong, Hyung-soo Kim, Hyung-Jo Jung

**Affiliations:** Department of Civil and Environmental Engineering, KAIST, Daejeon 34141, Republic of Korea; kevin8633@kaist.ac.kr (J.S.); rlagudtn0934@kaist.ac.kr (H.-s.K.)

**Keywords:** construction management, safety in construction site, collision warning, CCTV, deep-learning-based object detection

## Abstract

According to data from the Ministry of Employment and Labor in Korea, a significant portion of fatal accidents on construction sites occur due to collisions between construction workers and equipment, with many of these collisions being attributed to worker negligence. This study introduces a method for accurately localizing construction equipment and workers on-site, delineating areas prone to collisions as ‘a danger area of a collision’, and defining collision risk states. Utilizing advanced deep learning models which specialize in object detection, video footage obtained from strategically placed closed-circuit television (CCTV) cameras across the construction site is analyzed. The positions of each detected object are determined using transformation or homography matrices representing the conversion relationship between a sufficiently flat reference plane and image coordinates. Additionally, ‘a danger area of a collision’ is proposed for evaluating equipment collision risk based on the moving equipment’s speed, and the validity of this area is verified. Through this, the paper presents a system designed to preemptively identify potential collision risks, particularly when workers are located within the ‘danger area of a collision’, thereby mitigating accident risks on construction sites.

## 1. Introduction

The construction industry is one of the most dangerous industries, as many accidents occur, and according to the Bureau of Labor Statistics [1], in the U.S., construction workers accounted for nearly 20% (1061) of all industry fatalities in 2019. According to the Ministry of Employment and Labor’s “Analysis of Industrial Accidents in 2020”, small- and medium-sized construction sites with fewer than 50 employees accounted for 72.1% of all accidents in the construction industry [2]. In particular, the total value of the construction industry per 10,000 individuals has increased as the contract amount of construction has decreased, and construction sites with a budget of less than USD 2 million account for more than 60 per cent of all construction accident fatalities. Nevertheless, a lack of funding for safety management and a shortage of labor for safety and management oversight still puts many workers at risk on small- and medium-sized construction sites.

According to the ‘2021 Industrial Accident Status Report’ published by the Korean Occupational Safety and Health Agency, there were 828 deaths and 102,278 injuries due to direct accidents in 2021. Of the total injuries, 26.3% (26,888) occurred in the construction industry, which also accounted for 50.4% (417) of the total fatalities, demonstrating that there was a high accident rate in the construction industry in 2021 [3]. Furthermore, the report noted that the number of occupational accidents caused by collisions with equipment in Korea increased by 9.5% year on year, and the number of collisions was on the rise from 2017 to 2021 [3,4,5,6,7].

Fatal collisions with construction equipment are a frequent occurrence on construction sites, and various studies have been conducted in an attempt to ensure the safety of construction sites [8,9,10]. According to OSHA, the main cause of collisions with construction equipment is equipment operator inattention [11]. For this reason, preemptive alarms are recommended when there is a ‘collision danger state’. With safety and anti-theft monitoring systems in place at most construction sites, safety management strategies are being developed to prevent potential collisions between workers and construction equipment using computer-vision-based techniques.

Collisions involving construction equipment, commonly referred to as “collisions”, require significant attention to safety management due to their high frequency and potential for severe injuries. These accidents often occur when workers enter the blind spots of construction equipment operators, highlighting the need for safety protocols such as the deployment of signalers during the operation of construction equipment. Research is currently underway to detect proximity to construction equipment and identify direct hazards using cameras or sensors attached to the equipment [12]. However, given the cost of implementing such measures across all equipment on small- and medium-sized construction sites, a tailored safety management strategy specific to these sites is required.

This paper presents a feasible computer-vision-based system for detecting collision risk between construction equipment and workers by estimating the positions of construction equipment and people on small- and medium-sized sites using fixed CCTV images and defining the danger state of a collision based on the estimated positions. The proposed system includes various image processing techniques such as object detection, object tracking, and projection transformation to ensure the stability of object detection and analyze the spatial and temporal features of CCTV image data. We propose a definition of a danger area of a collision and a system for locating objects using a single fixed CCTV to determine the collision risk in advance. We define an area where the risk of collision increases when a worker enters and exits a danger area of a collision and verify the feasibility of this via a simulation using a virtual space implemented through a game engine. We also evaluate the feasibility of using a particular method for estimating the location of construction equipment and workers on small- and medium-sized sites using fixed CCTV images through a virtual space and verify the feasibility of defining a collision risk state according to the movement of workers and construction equipment via a field application.

The remainder of this paper is organized as follows. Section 2 discusses related work on the detection of collision between workers and construction equipment and the position estimation of objects on a construction site based on video, as well as the limitations of these studies. Section 3 introduces the system configurations, the models used for each configuration, as well as the techniques used. Section 4 evaluates the performance of the components of the system in virtual space and analyzes the validity of the definition of collision risk situations. Section 5 analyzes the overall performance of the system by applying the system in virtual space and in the real world. Finally, Section 6 presents our conclusions, limitations, and future research.

## 2. Related Works

Many studies propose the use of real-time monitoring systems to detect and alert operators to hazardous conditions around construction equipment in order to mitigate operator error and prevent accidents caused by collisions between operators and equipment. This section introduces the real-time research focused on locating the objects required to prevent collisions between workers and construction equipment and the danger of collision detection methodology based on the position of objects. We examine existing research on locating techniques using various sensors and cameras, as well as the methodologies leveraging these techniques to define and identify the danger state of collisions based on construction equipment and worker positions. This review emphasizes the strengths and limitations of each technique and methodology.

### 2.1. Locating the Object

Recent technological advances have greatly enhanced construction safety research. Tools like GNSS have gained prominence, while real-time location systems (RTLSs) are becoming increasingly popular. These RTLSs encompass a range of technologies, including RFID, GPS, UWB, WLAN, ultrasonic and infrared sensors, as well as vision-based analytics [12].

Ding et al. and Lu et al. proposed a location detection system for construction workers and equipment based on RFID technology [13,14]. These RFID tags are passive, making them advantageous due to their portability; they can be easily attached to various construction equipment or workers without the need for an independent power source. However, there are drawbacks. Fixed RFID readers must be strategically placed around the construction site. While this is cost-effective, it is best suited for smaller construction sites. The installation and operation of these RFID readers can take up a lot of space and be costly. Additionally, workers must always carry the RFID tags for the system to function.

Systems utilizing GPS can work with four or more satellites to determine the location of workers or equipment [15]. Unlike RFID systems, which do not require a separate battery to operate, GPS sensors are active sensors that require a separate power supply for operation and have the limitation of being less portable. In particular, they can only identify the location of the worker using the GPS sensor.

RTLS methods using sensors such as RFID and GPS are limited in the way that workers need to carry separate devices in order to be located on construction sites. On the other hand, vision-based analysis methods have the advantage that the target object does not need to carry any devices. This is a great advantage for use on small sites. Son et al. presented a method for localizing workers at a construction site through camera footage installed on construction equipment [16]. When using this method, workers are detected by the installed cameras, and their location is determined by estimating the distance between the workers and the equipment based on the angle at which the cameras are installed. This method has the advantage of identifying workers who do not have separate sensors such as RFID or GPS, but it has the limitation that separate cameras must be installed on the construction equipment in multiple directions.

Kim et al. presented a computer-vision-based methodology for the position estimation of construction equipment and workers using unmanned aerial vehicles (UAVs) [17]. A methodology was developed to measure position and distance using 2D images with the Yolo-v3 object detection model. By converting UAV images to orthophoto images using a quadrature conversion technique, it has the advantage of being applied to all construction workers and equipment captured in UAV images without additional sensors, but it has the limitation that it can only be used in environments where UAV images are available and is difficult to operate on small sites.

Luo et al. and Zhang et al. proposed a video-based method for estimating the distance between construction machinery and workers [18,19]. Luo et al. proposed a method to detect collision risk by estimating the distance between construction equipment and workers using object detection and quadrature transformation techniques based on images collected on construction sites. However, there is a limitation that they did not evaluate the accuracy of estimating the exact position of the construction equipment. Zhang et al. estimated the distance between the equipment and the worker based on pixels and used a fuzzy method to determine the collision risk, but this method has the limitation that it does not take into account the distortion that occurs as the picture moves to the periphery. In addition, the evaluation is performed only on a frame-by-frame basis, which does not take into account changes in the risk area due to the movement of the construction equipment.

### 2.2. Detecting the Collision Danger State

On existing construction sites, the identification of the danger state of a collision primarily hinges on two methodologies: the method predicting future paths based on the worker’s previous trajectory and determining a dangerous situation when the worker enters the danger zone of collision, and the method defining a dangerous situation based on the proximity between the worker and construction equipment to ascertain the danger state of a collision.

Zhu et al. introduced a methodology utilizing multiple video cameras to gauge the positions of machines and operators [10]. When using this method, these data are then processed through an innovative Kalman filter to predict future locations. This filter bases its predictions on the past movements of the operator or machine.

Rashid et al. employed two trajectory prediction models: multinomial regression and the hidden Markov model (HMM) to forecast the operator’s trajectory and detect potential hazards [20,21]. However, these models do not consider that the hazard area’s dimensions and location can vary depending on the movement and size of the construction equipment, rendering them less suitable for collision risk detection.

Zhang et al. utilized a machine-learning-based object detection model to identify the basket and operator of an excavator [19]. They then employed fuzzy inference to extract central pixel coordinates and congestion to gauge collision risk. This method, however, does not factor in the machine’s size and movement. It estimates the distance between the machine and operator based solely on pixels, which can lead to inaccuracies.

Conversely, Luo et al. suggested a technique that establishes the danger radius centered on the equipment’s center [18]. This is derived from the bounding box of the construction equipment’s object detection results. Yet, this approach does not account for equipment movement. Additionally, the method’s estimation of the equipment’s center of gravity may be skewed due to camera positioning and angles. This limitation hampers the accurate evaluation of the construction equipment’s exact position, potentially affecting collision risk assessment.

## 3. Methodology

As mentioned in the Introduction, accidents on construction sites are concentrated on small- and medium-sized sites. To address these issues, the Seoul Government enacted the ‘10 Measures to Strengthen Safety Management at Small and Medium-sized Private Construction Sites’, which extends the mandatory installation of CCTV to sites of $10,000\;m^{2}$ or less, which was previously only applicable to sites of $10,000\;m^{2}$ or more. However, there is a problem: it is difficult to install expensive equipment such as stereo cameras or multiple cameras on small- and medium-sized construction sites due to financial burdens. In this paper, we propose a collision detection technology between construction equipment and workers by utilizing a single CCTV with relatively inexpensive HD (1280×720) and FHD (1920×1080) resolutions. In particular, we aim to improve the efficiency of construction safety management and contribute to accident prevention by studying and implementing collision detection technology within the shooting range through CCTV shooting at outdoor construction sites where construction equipment is actively moved to transport building materials.

The system designed to reduce the risk of collision accidents works by applying an object detection model to the image data collected from the CCTV, as shown in Figure 1. Therefore, the accuracy of the system is highly dependent on the performance of the object detection model. To evaluate the accuracy of the locating sub-system and detecting the dangerous state of a collision sub-system, it is necessary to verify that the results of the object detection model match the ground truth. Additionally, before validating the performance of individual technologies, it is essential to establish a site where CCTV image data can be collected. This site will collect frame-by-frame data on the locations of construction equipment and workers, as well as the collisions between them. However, it is a difficult task to create realistic collision scenarios on a construction site and identify the exact location of each equipment and worker. Therefore, in this study, a virtual environment is built using the Unity game engine, and the performance of each component technology is verified via a simulation in this virtual environment using images that understand the ground truth, such as the location of construction equipment and workers, the bounding box of objects, and whether they collide. The verified system was then applied to the CCTV video data collected from actual construction sites to evaluate the performance of the entire system using object detection technology.

Figure 1 illustrates the interconnection structure among subsystems within a system designed to detect potential collision risk conditions between construction equipment and workers, utilizing a single fixed CCTV installed at a construction site. Figure 2 delineates the specific flow diagram of each subsystem as well as the comprehensive workflow of the overall system. The system comprises two subsystems: the locating system and the detecting system for the danger state of a collision. This section describes the methodology for estimating the location of construction equipment and workers based on a single fixed CCTV and the definition of the danger state of a collision.

### 3.1. The Locating System

The locating system applies a deep-learning-based object recognition model to CCTV images to detect workers and construction equipment at a construction site; it estimates the location of the equipment and workers via quadrature transformation. The surface of the construction site is assumed to be flat, and all the construction equipment and workers are assumed to be moving on the surface. At this time, the structures installed on the construction site are used as the control point.

#### 3.1.1. Deep-Learning-Based Object Detection

In order to deploy the system at a construction site, it is necessary to apply an object detection model to the video footage collected from the CCTV installed on the site. At this juncture, for the effective recognition of collision risk conditions using CCTV, it is critical that the employed object detection model accurately identifies construction equipment and workers in real-time within each frame of the video, enabling real-time risk detection and assessment. In general, object detection using deep learning techniques can be divided into three main categories: two-stage, one-stage, and transformer-based detector [22].

A two-stage detector is a model that has a separate stage for locating objects and a separate stage for classifying objects. In this way, the model finds regions in the image where an arbitrary number of objects are estimated to be present in the first stage and then classifies them in the second stage. Because the system has two stages, it takes longer to perform the task and has a more complex architecture.

On the other hand, one-stage detectors use dense sampling to locate objects and classify them simultaneously. This simplifies the traditional two-stage method and speeds up the process. 

Transformer models are a way to detect objects by applying transformers from the field of NLP to the field of computer vision. Transformers are techniques that mathematically find patterns between elements in data. In many areas of computer vision, transformer models have been shown to perform well, but they require more data to train than traditional models.

In order to detect the danger state of a collision using CCTV, it is necessary to recognize objects in real time, frame-by-frame. In addition, the accuracy of object detection must be sufficiently reliable. Therefore, in this paper, the YOLO-v5 model, a fast and highly accurate one-stage model, is used as an object detection model for collision risk detection. The small-scale model of the YOLO-v5 model (hereinafter referred to as ‘yolov5s’) was used. The dataset consists of 12,298 fixed CCTV camera images collected from seven small- and medium-sized construction sites and 9392 web crawl images, as shown in Figure 3. To train the object detection model, 80% of the total collected images are used for the training set, and 20% are used for the validation set. All images from small- and medium-sized construction sites were collected using 2 MP resolution cameras and resized to 640 during training. The object detection model was performed on an Intel i9-10980XE CPU, 256 GB RAM, and NVIDIA GeForce RTX 3090 GPU. Figure 4 details the performance of the model.

#### 3.1.2. Projective Geometry

According to projective geometry, when an arbitrary plane in three-dimensional space is projected onto another plane centered on an arbitrary point, certain transformation relations exist between the projected corresponding points. On a construction site, construction equipment and workers can be assumed to move on a surface. If the surface of the construction site is not very uneven, assuming that the surface is flat, the image captured by CCTV can be considered to be a quadrature transformation of the surface, and the quadrature transformation matrix (hereinafter referred to as the homography matrix) can be obtained by the direct linear transform (DLT) algorithm using singular value decomposition (SVD) with four or more control points [23]. The safety of the homography matrix due to errors such as human error and rounding error varies greatly depending on the conditions for selecting the control points. The selection of control points is subject to the following conditions: four or more control points not on the same line; two or more control points that do not span a line for any line; set the center of the on-camera landmark to the pixel coordinates of the control points.

Table 1 shows the error in location estimation by quadrature transformation when the control points are selected according to the conditions and when they are not according to the conditions for a reference point whose location is known. The average location error (ALE) is calculated using the following equation:(1)$$ALE=\frac{1}{n}\sum_{i=1}^{n}\sqrt{{\left(x_{gt}-x_{e}\right)}^{2}+{\left(y_{gt}-y_{e}\right)}^{2}}$$
where $x_{gt}$ and $y_{gt}$ are the coordinates of ground truth (m), and $x_{e}$ and $y_{e}$ are the estimated coordinates (m).

The error (ALE) was 36.95 (m) for 38,400 reference points when the condition was not met, and the error (ALE) was 0.07 (m) for 24,893 reference points when the condition was met.

#### 3.1.3. Extracting Location Representatives

In this paper, an object’s position is defined as the central point of the bottom surface of its minimal bounding box. However, as depicted in Figure 5, the projected transformation result of the bounding box’s center in the image exhibits significant deviation from the object’s actual position. Kim et al. captured construction site images from an elevated angle using a UAV and rectified the image into an orthographic view via a projection transformation, thereby approximating the object’s center of gravity as its location [17]. In contrast, construction site CCTVs typically capture images from a more oblique angle than UAVs. This often results in extensive shadowing from objects and pronounced distortion, as illustrated in Figure 6. In contrast to typical imaging perspectives, Figure 6a presents the converted result of an image captured at a higher angle, while Figure 6b illustrates the same for a lower angle. The sky-blue arrow highlights the discrepancy between the true focal point and the bounding box’s central point post-conversion. Notably, in images taken from a lower angle, this discrepancy is magnified due to distortions from shading. Given these challenges, it becomes imperative to identify a representative point that can precisely estimate an object’s location, considering both the bounding box and object class.

The human object, such as a worker, can be approximated by a thin, long cube. In this case, the representative point from which the worker’s position is estimated is located on the vertical centerline of the worker’s bounding box. However, the vertical position of the representative point is uncertain. As shown in Figure 7, we assume that the representative point is located at αh relative to the top of the bounding box, where α denotes the position coefficient. The operator can see that the value of α is almost constant.

On the other hand, as shown in Figure 8, the position of the construction equipment’s representative points varies greatly depending on the position and posture of the equipment. Therefore, we propose a methodology to estimate the position representative points of construction equipment based on the VGG-19 model, which is often used in image preprocessing in skeleton detection models [24]. The VGG-19 model is used to infer image features through a CNN layer and is commonly used in image preprocessing. The structure of the VGG-19 model is shown in Figure 9.

### 3.2. The Definition for the Danger State of Collision

In this paper, the dangerous state of collision is defined as the state in which the worker is within the collision danger area. In order to define the collision danger area, the driving range of the construction equipment must be considered. In particular, since collisions on construction sites are often caused by the inattention of the machine operator, it is necessary to consider the stopping distance caused by the machine’s sudden braking. Stopping distance is the distance travelled by the machine from the driver’s perception of danger to the immediate reaction to stop. The stopping distance is divided into reaction distance and braking distance. The reaction distance refers to the interval between the driver’s perception of danger, and the braking distance refers to the deceleration period after the brakes are applied. Assuming that the deceleration of the deceleration interval is constant, the stopping distance ($d_{s}$) is expressed as:(2)$$d_{s}=Vt_{r}+V^{2}/2J$$
where $d_{s}$ is the stopping distance (m), V is the velocity of a vehicle (m/s), $t_{r}$ is the reaction time, (s) and J is the braking acceleration $\left(m/s^{2}\right)$. 

In practice, the reaction time is defined as approximately 1 s. On the construction site, the speed of construction equipment is limited to no more than 20 km/h by the local rule on occupational safety and health standards [25]. In addition, the braking distance can be assumed to be zero because the braking decoration of construction equipment is large enough. Therefore, the stopping distance ($d_{s}$) is expressed as:(3)$$d_{s}=Vt_{r}$$

In this paper, for stationary construction equipment, we define the construction equipment collision danger area as a circle centered on the position of the equipment. The radius of this circle is calculated by adding the size of the equipment to the maximum walking distance that the operator can travel after perceiving the hazard until the operator immediately stops moving. Using a typical human walking speed of 1 m/s and a reaction time of 1 s, we determined the maximum walking distance of a worker to be 1 m by multiplying their walking speed by their reaction time [26]. For moving construction equipment, we define the “danger area of collision” as the shape shown in Figure 10, which is a combination of a semi-ellipse with an extended radius that includes the stopping distance and a semi-ellipse, considering the size of the equipment and the stopping distance in the direction of movement.

a is the semi-minor axis of the ellipse and radius of the circle, and b is the semi-major axis of the ellipse:(4)$$a=S+V_{w}t_{r},\;b=S+{(V}_{w}+V)t_{r}$$
where S is the size of equipment (m), $t_{r}$ is the reaction time (s), and $V_{w}$ is the worker’s walking speed (m/s).

## 4. Performance Evaluation of Sub-Systems

Before validating the performance of individual technologies, there is a need to construct a site for collecting CCTV images to verify the technology. This site would also gather frame-by-frame data on the positioning of construction equipment and workers, as well as any collisions between them. However, as outlined in the methodology section, the creation of actual collision scenarios on construction sites and capturing the precise location and interactions between equipment and workers in each frame pose significant challenges. Therefore, we have opted to initially validate each component technology through simulations in a virtual environment crafted using the Unity game engine. This preliminary verification will be followed by field verification on actual construction sites to validate the system’s efficacy.

The virtual construction site features an L-shaped building model spanning 18 floors. It is presumed that all construction equipment and workers operate on the ground within a delineated boundary (represented by a black line). Figure 11 illustrates the shapes of the building model and the ground area. Workers are animated to move randomly across the ground, while construction equipment is animated based on its specific characteristics, allowing it to move around the site. The camera’s interior orientation parameters (IOP: including sensor size and focal length) and exterior orientation parameters (EOP: encompassing height, roll, and tilt) are detailed in Table 2.

### 4.1. The Locating System

The locating system determines the location of objects using bounding boxes derived from object detection and computes their position through quaternion transformation. A homography matrix is generated using landmarks selected based on the criteria detailed in Section 3.1.2.

Given that workers can be represented as thin, elongated cuboids, their representative points are nearly identical within the bounding box. We implemented this method on 25,200 images capturing workers moving randomly in a virtual space. Figure 12 displays the positional error relative to α. With α=0.914, the error is minimized to 0.0762 m, as shown in Figure 13. This accuracy surpasses the UAV-based method proposed by Kim et al.

However, the reference points for construction equipment change significantly based on their position and orientation. To estimate these points, we employed the VGG-19 model. Images cropped around the equipment’s bounding box, created in Unity’s virtual space, served as inputs, producing α and β as outputs. We compiled a dataset of 56,104 entries covering three types of construction machinery (truck, excavator, and forklift) and trained the VGG-19 model with a training-to-validation set ratio of 4:1. Figure 14 shows the model’s loss.

We implemented this method on 24,900 images (bongo: 9000, excavator: 7800, and forklift: 8100) capturing equipment moving randomly in a virtual space. Table 3 presents the locating error for construction equipment using the VGG-19 model. As Figure 15 illustrates, errors increase when equipment is partially visible within the camera’s frame. However, when fully within view, as presented in Table 4, accuracy improves. The UAV method has positional errors of 0.31 m and 0.37 m for trucks and excavators, respectively. Thus, our method, which uses low-angle CCTV shots with significant shading from equipment, offers comparable or better accuracy. Figure 16a further confirms the precise position estimation.

Nevertheless, when observing the movement of construction equipment, variations in direction and size arise from localization errors, as depicted in Figure 16b. Such discrepancies are anticipated to influence “a danger area of a collision”, factoring in the equipment’s speed, leading to potential misdetections.

### 4.2. The Danger State of Collision

Shin et al. assessed collision risk by categorizing risk levels based on the proximity between a worker and construction equipment [26]. The highest collision risk level is designated when a circle, centered on the worker with a radius of 1 m, overlaps with construction equipment. Building upon this definition, this paper identifies such an overlapping scenario as a ‘collision state’. The validity of this definition of “a danger area of a collision” is confirmed by comparing the presence of workers in the area defined in Figure 10 with the actual “collision state” This is carried out using a 10 s simulation video in a virtual space.

The outcomes, sourced from 208 videos (safe: 174, danger: 34), are tabulated in Table 5. ‘True dangerous’ video refers to one where a ‘collision state’ occurred at least once, while ‘True safe’ is one without any ‘collision state’. ‘Detected dangerous’ indicates a video where the defined ‘danger state of collision’ was observed at least once, and ‘Detected safe’ is its counterpart.

As a result of the analysis, precision was calculated as 82.9%, and recall was calculated as 100%. Precision is the ratio of what is actually “true duality” among what is classified as “detected duality”, and recall is the ratio of what is actually “true duality” classified as “detected duality”. Notably, there were no instances of ‘false safety’, meaning situations where a “True dangerous” was detected as safe. As seen in Figure 17, of the seven identified ‘false dangers’, six (Figure 17a–f) were scenarios where a worker intersected the movement path of construction equipment, and one (Figure 17g) was when a worker was in the path of equipment rotation. While these seven situations do not align with the ‘collision state’ definition, they undeniably represent hazardous conditions. This underlines the reasonability of the ‘collision danger state’ definition. When a true danger was correctly detected, it was predicted an average of 28.53 frames (about 0.951 s) earlier at 30 fps.

## 5. Results

The system was assessed using videos created within a virtual space implemented through a game engine (Figure 11) and real-world videos obtained from a construction site, focusing on the identification of the “dangerous state of a collision.” Prior to the system’s application, each of the 10 s videos was classified into “True dangerous” or “True safe” categories.

Within the virtual setting, a total of 82 videos were produced: 11 were designated as “True dangerous” and 71 as “True safe.” In this simulated environment, both workers and construction equipment (comprising trucks, forklifts, and excavators) exhibited random movements. The respective lengths of these machines were 5.5 m, 11.4 m, and 5.3 m, and they operated at speeds not exceeding 2 m/s.

As a result of applying the system to a virtual space video, the precision was calculated to be 64.7%, and the recall was calculated to be 100%. As indicated by Figure 18, the system successfully detected all the “True dangerous” videos. However, Table 6 reports six instances where “False dangerous” judgments occurred, suggesting the system’s propensity to overestimate potential risks.

As a result of applying the system to a real construction site video, the precision was calculated to be 73.3%, and the recall was calculated to be 100%. The classification outcomes for videos retrieved from an actual construction site are detailed in Figure 19. From the 25 videos sourced, 11 were categorized as “True dangerous” and 14 as “True safe”. According to Table 7, while there were no identifications of “false safe”, four videos were inaccurately judged as “false dangerous”, underscoring a consistent pattern observed in the virtual testing.

The system demonstrated a tendency to overemphasize potential hazards. Figure 16b displays an enlarged segment from the results of the locating equipment shown in Figure 16, specifically focusing on areas where the X-coordinates range from 12 to 16 and the Y-coordinates from 3.5 to 4.5. The blue arrow indicates the actual movement direction and speed using the system, while the yellow arrow represents the estimated direction and speed of movement. Due to errors in location estimation, deviations from the actual movement direction and speed were observed even within short time intervals. These discrepancies appear to induce an overestimation when determining the ‘danger area of a collision’ based on equipment speed, consequently leading to ‘false dangerous’ judgments.

## 6. Conclusions

In this study, a new method is proposed for accurately localizing workers and machines on construction sites using fixed CCTV, along with a “danger area of a collision” to assess the danger of a collision. The accuracy of the location estimation system was assessed using average location error (ALE), registering 0.0762 m for workers, 0.1934 m for trucks, 0.423 m for excavators, and 0.2740 m for forklifts. Additionally, the “collision state” identified using the “danger area of a collision” demonstrated high accuracy in real-world risk situations, with a precision of 82.9% and a recall rate of 100%. The system exhibited a precision of 64.7% and a recall rate of 100% in virtual space, and 73.3% precision and 100% recall on a real construction site. These results suggest the feasibility of developing a system capable of monitoring potential collision risks without the need for specialized equipment such as GPS sensors or UAVs. However, during system validation, we encountered “false danger zones” in both simulated and real-world environments. 

These inaccuracies were traced back to misalignments in machine and human positioning and complex movement dynamics. The crux of the issue lay in the miscalculation of machine trajectory and speed, distorting the defined “danger area of a collision”, particularly when speed estimates were inaccurate. These misalignments often resulted in ‘false danger zones.’ To mitigate this, enhancements in machine positioning accuracy and movement prediction are imperative. While corrective measures can fine-tune equipment, movement is estimated if positioning errors exhibit a consistent pattern; the context-dependent nature of machine positioning errors necessitates further research in order to improve positioning accuracy. Furthermore, there is a need to expand the definition of the ‘danger area of a collision’. Especially during the system evaluation in virtual space, the majority of false danger instances were observed due to an overestimation of collision risk during the rotation of construction equipment. This overestimation stemmed from defining the ‘danger area of a collision’ solely based on the equipment’s speed, without considering its rotation. Therefore, subsequent research is needed to identify the most optimized model considering the shape and rotation of the equipment, building upon the ‘danger area of a collision’ defined in this study that considers the speed of construction equipment.

## Figures and Tables

**Figure 1 sensors-23-08371-f001:**
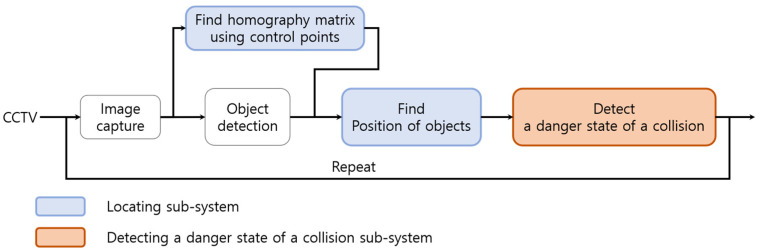
The algorithm for detecting the danger state of a collision.

**Figure 2 sensors-23-08371-f002:**
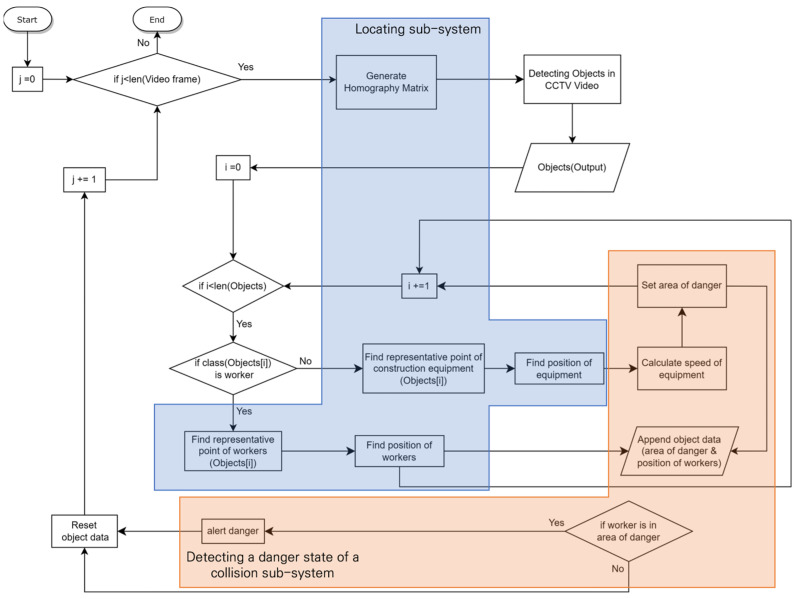
The flowchart of an algorithm for detecting the danger state of a collision.

**Figure 3 sensors-23-08371-f003:**
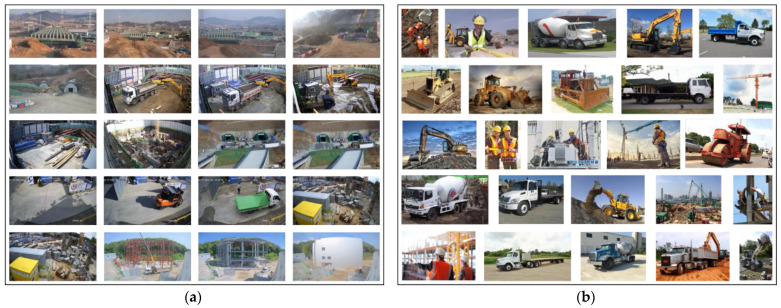
The dataset for object detection: (**a**) construction site; (**b**) web crawling.

**Figure 4 sensors-23-08371-f004:**
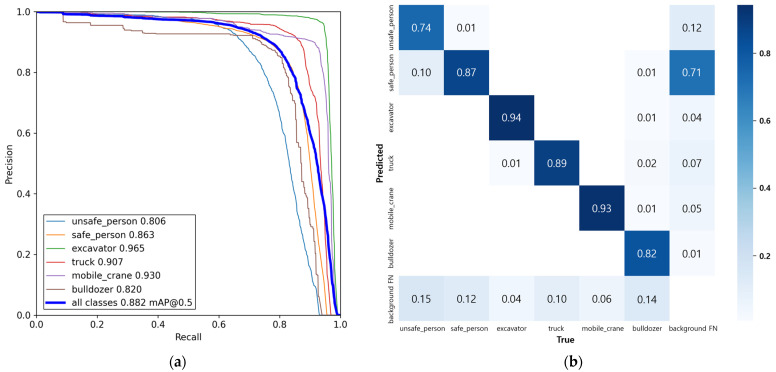
The performance of the YOLOv5L model: (**a**) PR curve; (**b**) confusion matrix.

**Figure 5 sensors-23-08371-f005:**
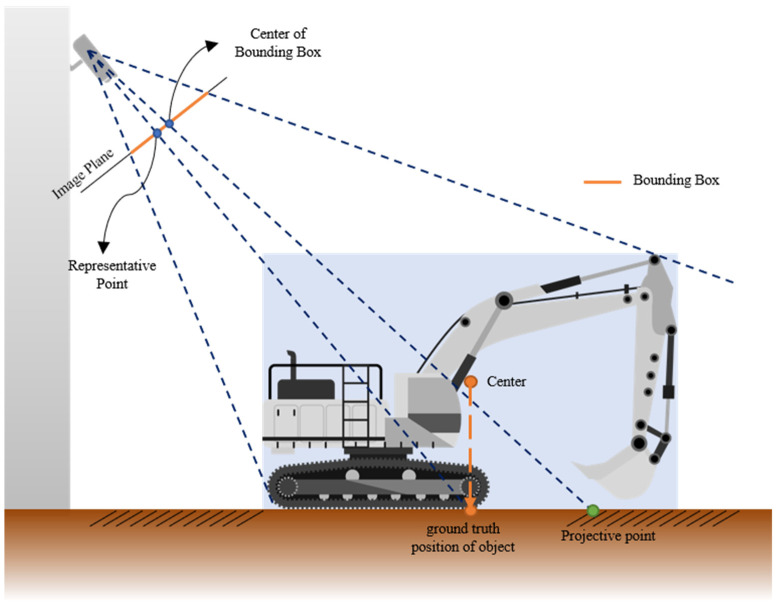
The projective point of center of the bounding box.

**Figure 6 sensors-23-08371-f006:**
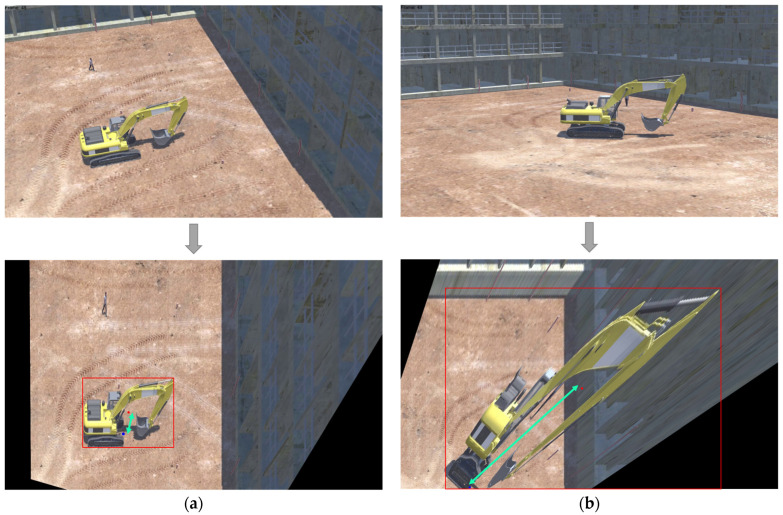
The result of image correction using the homography matrix based on shooting angle; red dot; center of bounding box; blue dot; center of the object; (**a**) at a high angle; (**b**) at a low angle.

**Figure 7 sensors-23-08371-f007:**
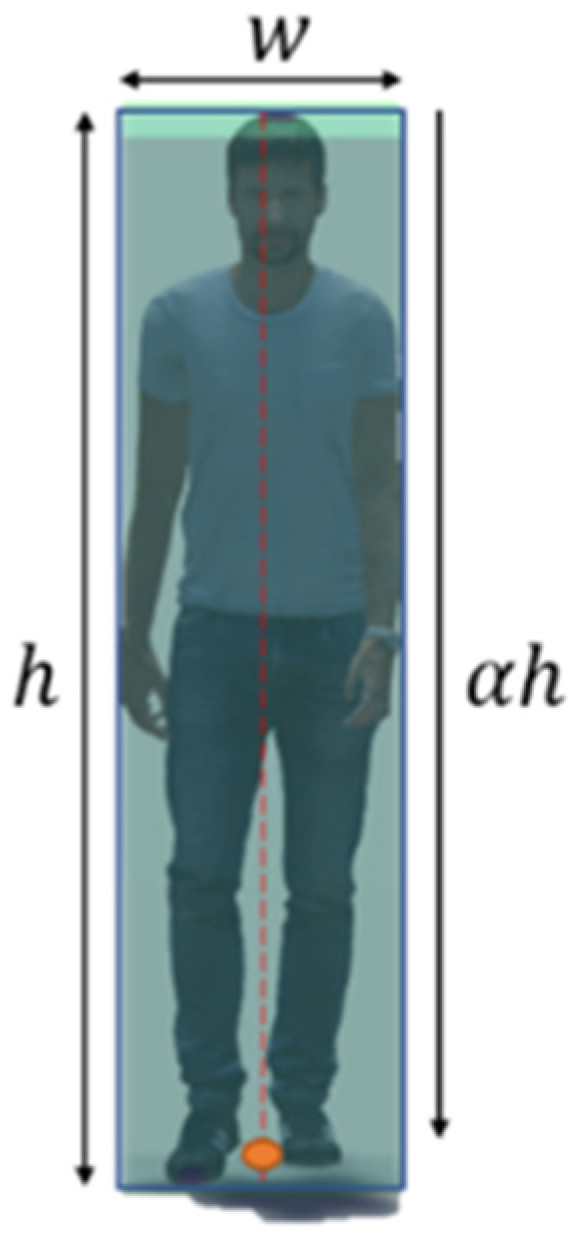
The bounding box and representative of a human object.

**Figure 8 sensors-23-08371-f008:**
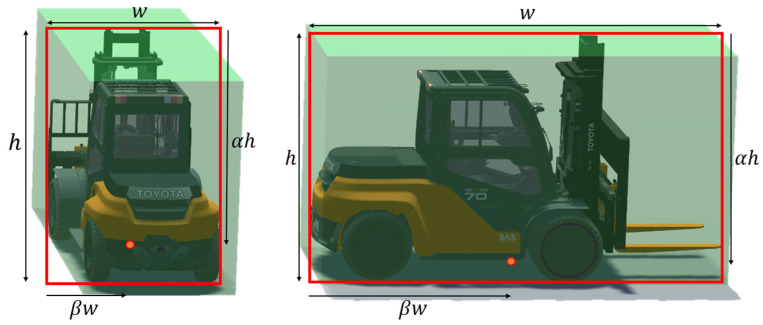
The bounding box and representative of equipment.

**Figure 9 sensors-23-08371-f009:**
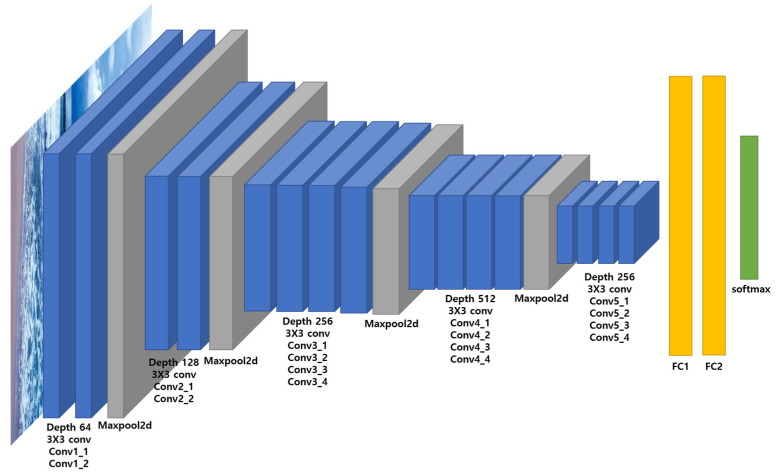
The architecture of VGG-19.

**Figure 10 sensors-23-08371-f010:**
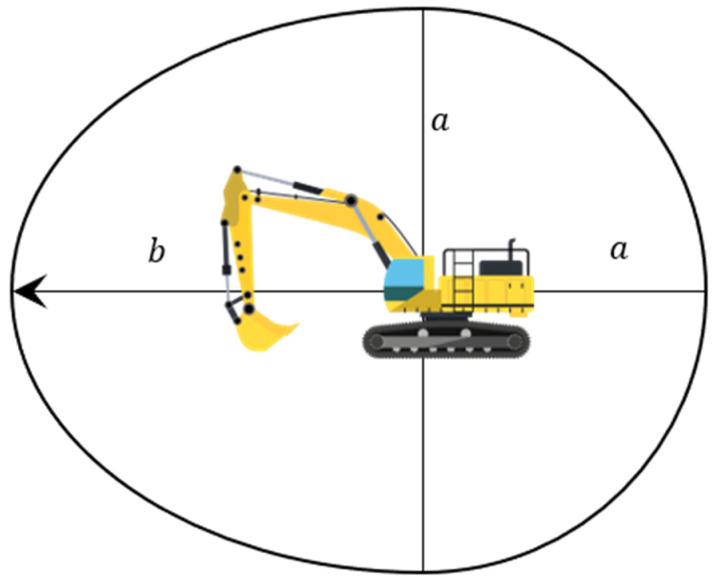
The danger area of the collision.

**Figure 11 sensors-23-08371-f011:**
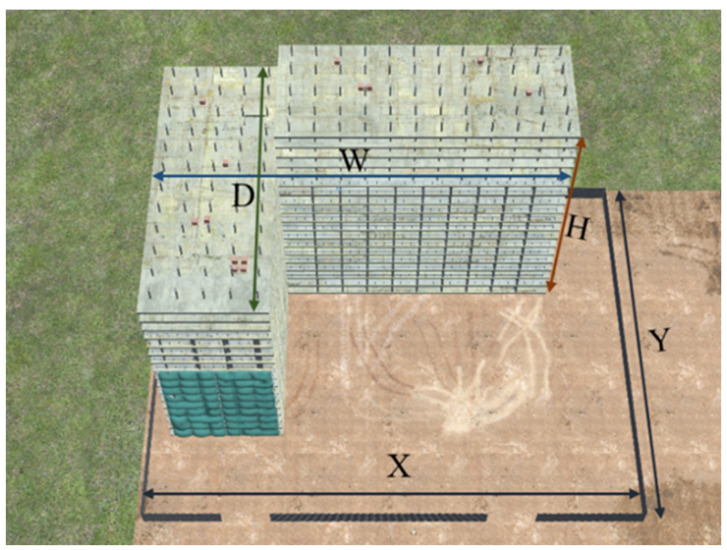
Building model (W×D×H) and the ground surface (X×Y).

**Figure 12 sensors-23-08371-f012:**
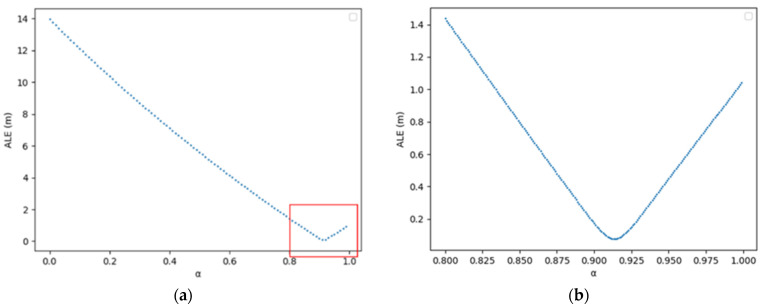
Error (ALE) of locating system for a worker: (**a**) original; (**b**) enlarged.

**Figure 13 sensors-23-08371-f013:**
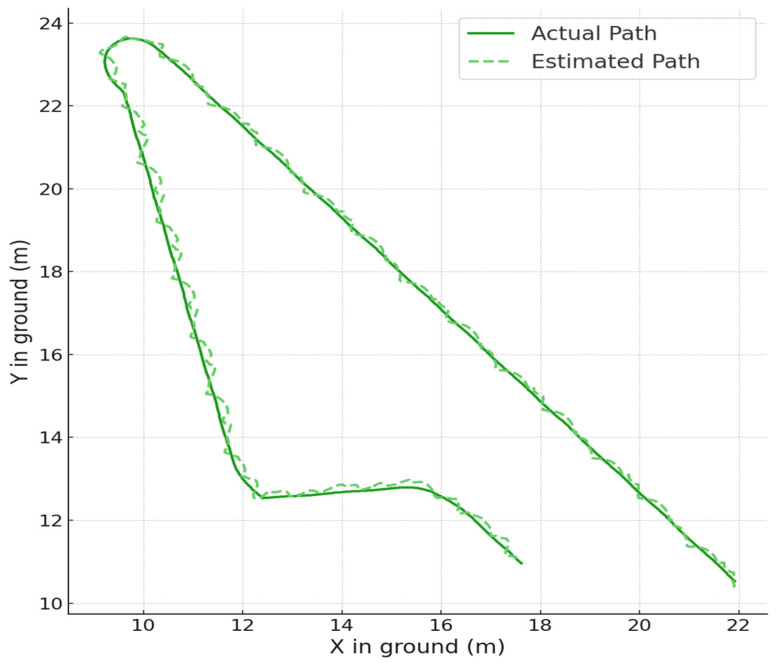
Result of locating a worker.

**Figure 14 sensors-23-08371-f014:**
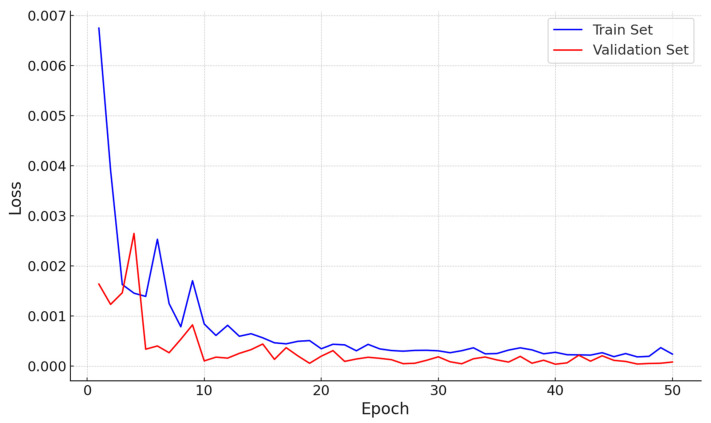
Training and validation loss.

**Figure 15 sensors-23-08371-f015:**
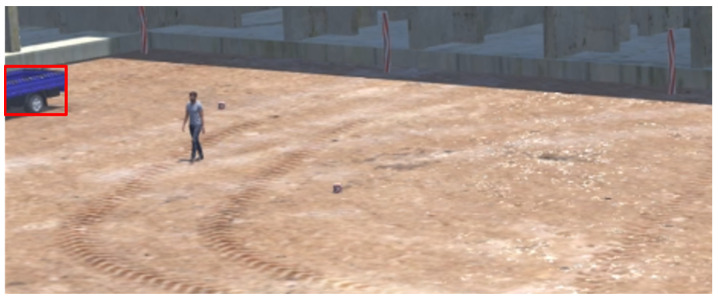
Construction equipment partially out of camera view.

**Figure 16 sensors-23-08371-f016:**
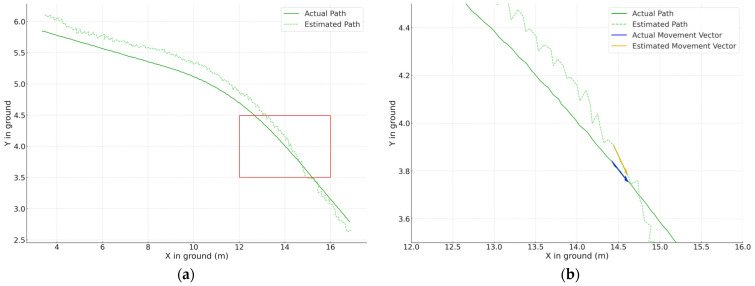
Result of locating equipment (excavator): (**a**) original; (**b**) enlarged (red box).

**Figure 17 sensors-23-08371-f017:**
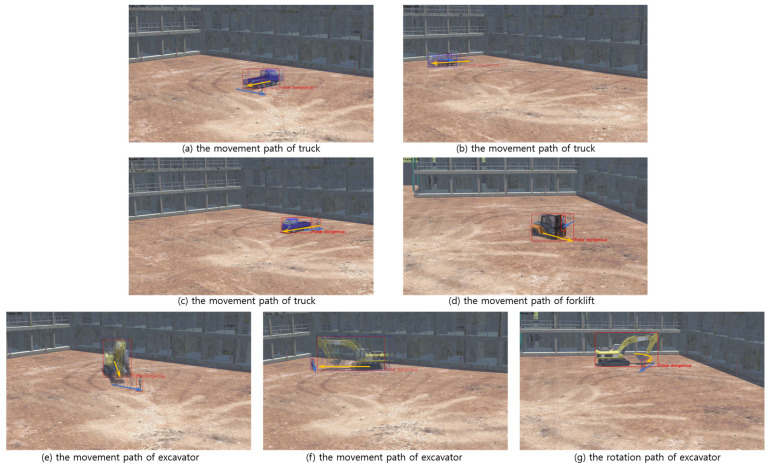
The images of “false dangerous”; yellow arrows: movement of equipment; blue arrows; movement of workers.

**Figure 18 sensors-23-08371-f018:**
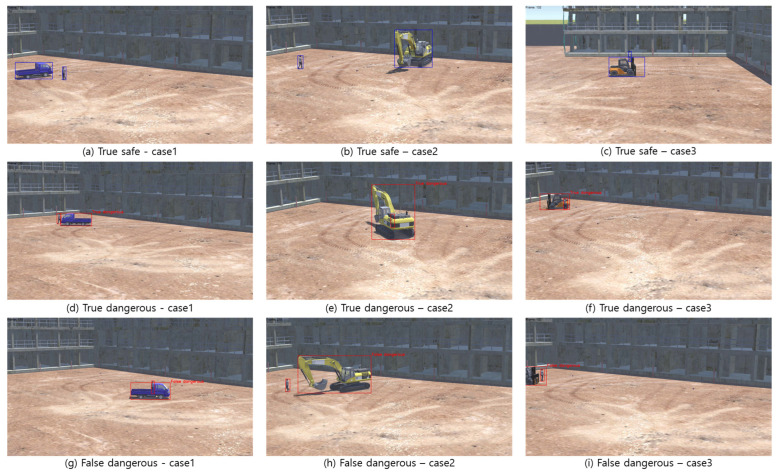
The result of the detection system in virtual space; blue box: detected safe state; red box: detected dangerous state.

**Figure 19 sensors-23-08371-f019:**
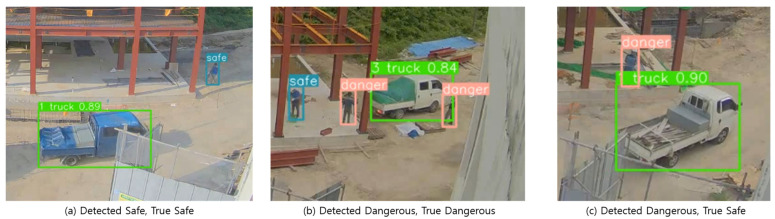
The result of the detection system on a real construction site.

**Table 1 sensors-23-08371-t001:** Locating error based on condition compliance.

	Number of Reference Points	ALE (m)
Condition not met	38,400	36.95
Condition met	24,893	0.07

**Table 2 sensors-23-08371-t002:** The range of IOP and EOP of CCTV.

Definition	Symbol	Range	Unit
Height	H	6–12	m
Tilt	θ	5–20	°
Roll	ψ	−1.5–1.5	°
Focal length	f	13.5	mm
Horizontal size of the image sensor	$H_{sensor}$	24.89	mm
Vertical size of the image sensor	$L_{sensor}$	18.66	mm
Resolution	H×L	1920×1080	-

**Table 3 sensors-23-08371-t003:** The locating error for construction equipment.

Type of Equipment	ALE (m)
Truck	0.2544
Excavator	0.5473
Forklift	0.3208

**Table 4 sensors-23-08371-t004:** The locating error for construction equipment except the frame when construction equipment is partially out of camera view.

Type of Equipment	ALE (m)
Truck	0.1934
Excavator	0.423
Forklift	0.2740

**Table 5 sensors-23-08371-t005:** Evaluate the validity of the definition of “collision state”.

	True Dangerous	True Safe
Detected dangerous	34	7
Detected safe	0	167

**Table 6 sensors-23-08371-t006:** Performance of the detection system in virtual space.

	True Dangerous	True Safe
Detected dangerous	11	6
Detected safe	0	65

**Table 7 sensors-23-08371-t007:** Performance of the detection system on a real construction site.

	True Dangerous	True Safe
Detected dangerous	11	4
Detected safe	0	10

## Data Availability

Not applicable.

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
