# Peer review of "The Detection System for a Danger State of a Collision between Construction Equipment and Workers Using Fixed CCTV on Construction Sites"

_sensors, 2023, doi:10.3390/s23208371_

Round 1

Reviewer 1 Report

This study aims to develop and validate a system to detect the potential collision risks between construction equipment and workers on-site by identifying their locations and defining the areas and state of collision. It is not clear how the authors have achieved this aim according to what has been reported in this study. The manuscript contains numerous unclear statements and grammatical errors. The following are a few of the questions and comments that need to be answered/addressed in the manuscript.

Line 17: Which speed is being referred to here?

Line 9: This sentence is not clear and needs to be revised.

Line 20: This part of the sentence needs to be revised.

Line 45: The authors should cite a few of the studies they are referencing to here.

Lines-46-47: This sentence should be properly/completely supported/cited.

Line 50-51: It is not clear what this part of the sentence means “…prevent potential collisions between computer vision-based workers and construction equipment.”

Line 170-173: The description of Figures 1 and 2 in these sentences is confusing and not clear.

Why was a single CCTV used? Where was the CCTV placed to be able to capture quality and accurate data needed for the study? These questions need to be answered and more details need to be provided in the manuscript about the data because the accuracy of results is highly dependent on the type and quality of data. Also, the practicality of the use of this single CCTV camera on a complex and dynamic construction site with constant interaction of heavy equipment and workers should be justified.

Additional information about the types of projects, and status or project phases of the construction sites used should be stated. This would impact the possibility of getting an adequate amount of quality data from the construction sites.

What are the types of images that make up the 12,298 and 9,392 images? Examples of these images should be included in the manuscript.

Lines 214-215: “…train set and valid set are divided 4:1” The authors should clarify the message they are trying to pass with this.

Lines 216-217: Training and testing results were performed…? This should be revised.

Line 218: The presentation of the performance of the model at this point appears to be too early and abrupt when an adequate amount of details have not been provided about the study.

The information in Table 1 need not be provided in a Table. It can be included in the manuscript as text.

Lines 250: Citation number is missing in Kim et al.

How is Figure 6 representative of a worker? No PPE including safety vest, helmet, or hard hat.

Line 327-328: The author should clearly state early in the manuscript that simulations in a virtual construction environment were used for validation in this study rather than real-life construction sites. Also, a more realistic virtual environment should have been developed and used for this study. The authors should provide a justification for not doing this.

The images in Figure 18(a) – (c) are too small and the distinction between the images has neither been described in the manuscript nor in the caption of the Figure.

Lines 443-444: The authors claimed to have created a pioneering method for pinpointing the locations of workers and machinery on construction sites, however, it is not clear how this has been done based on what has been reported in this manuscript.

In general, this manuscript seems to be disjointed and not well-structured. It is not clear what the contributions of this study are. The authors need to clearly report what has been done in this study in order to draw out its novelty and contributions. 

The manuscript contains numerous unclear statements and grammatical errors, therefore requires an extensive English language editing.

Reviewer 2 Report

Some comments:

- The manuscript looks like very interesting and it is a real new contribution to the topic - construction area.

- The main aim is clear. However, it must be included also in the Introduction section or even in a new section focused on Research Aim.

- Methodology is ok.

- The discussion of the results should be detailed with additional scientific contribution and comments related to others case studies in the construction area around the world.

- A paragraph considering future research works must be included at the end of conclusions section.

- Major revision. 

Moderate editing of English language required.

Reviewer 3 Report

This article outlines the hazardous areas of collisions by accurately locating the location between construction equipment and on-site workers, in order to determine the hazardous state of collisions and detect potential collision risks between construction equipment and on-site workers. This article introduces a system aimed at reducing the risk of accidents at construction sites by proactively identifying potential collision threats and determining whether workers are present in these designated collision hazard areas.

The reviewer has some comments and suggestions for the authors to further improve the paper.

1. Scientific contribution should be further clarified in the abstract and introduction section.

2. The abbreviation CCTV appears in the article abstract, which makes it difficult for non professionals to read;

3. Figures 2 and 3 have low accuracy and are difficult to accurately identify the content in the figure;

4. The conclusion contains uncertain words such as "seems to", lacking the rigor of research-oriented papers;

5. The conclusion should streamline the content of the research process, specifically introduce the main contributions, shortcomings, and future research directions of the research.

Extensive English editing is needed.

Round 2

Reviewer 1 Report

The reviewer would like to appreciate the authors for addressing all the comments raised and improving the quality of the paper. 

Reviewer 2 Report

Minor editing of English language required. I have no additional comments.

 Minor editing of English language required